# Research on dual-waterway cooling system of high-power-density permanent magnet synchronous machine

Yingying Xu, Xiaoqian Duan, Mingxia Xu*, Aochen Han, Wenyu Yu

School of Electrical Engineering, Dalian Jiaotong University, Dalian, Liaoning, China

* xumingxia@djtu.edu.cn

## Abstract

The issue of temperature rise and heat dissipation becomes crucial to enhancing motor performance as permanent magnet synchronous motors (PMSMs) for electric vehicles (EVs) advance toward high power densities and heat load densities. A unique frame-rotor dual-waterway cooling system is developed to address the heat dissipation problem of high-power-density permanent magnet synchronous machines (HPDPMSMs). The frame contains the outside water circuit, whilst the rotor support cylinder is fitted with an inner water circuit. The dual water circuits constitute a cooling circulation system via a water-cooled bearing chamber and the spinning shaft. Through thermodynamic calculations of the waterway, the structural characteristics are ascertained, and a fluid-structure interaction temperature field simulation model is developed. The efficiency of the dual waterway cooling structure is validated by comparing its cooling impact to that of the single waterway structure. The analysis of temperature distribution in the motor with a dual-waterway cooling structure, under different coolants, extreme operational conditions, and variable water velocities, validates the efficacy of the developed frame-rotor dual-waterway cooling system.

## Introduction

The national dual-carbon policy has garnered significant interest for EVs. The PMSM is extensively employed in EV drive systems owing to its remarkable efficiency, extensive operational speed range, straightforward architecture, and higher power factor [1]. As the driving range of EVs improves and installation space within the vehicle becomes constrained, the PMSM has progressively evolved towards higher power density [2]. Consequently, the escalation of motor losses and heat load density will directly result in an increase in temperature rise. Furthermore, the motor's integration into the vehicle chassis complicates heat dissipation, leading to a persistent increase in the motor's internal temperature, which accelerates insulation degradation and diminishes its operational lifespan. In extreme instances, it may result in

**Data availability statement:** All relevant data are within the manuscript.

**Funding:** This research was funded by the Science and Technology Bureau of Dalian, Liaoning Province, China, Grant number: 2023RQ061. The funders had no role in study design, data collection and analysis, decision to publish, or preparation of the manuscript.

**Competing interests:** The authors have declared that no competing interests exist.

irreversible demagnetization of permanent magnets, compromising the motor's power efficiency and dependability [3–5]. The motor's temperature increase substantially affects its performance and durability, while also indirectly impacting the operational condition of the fuel cell system [6]. Especially in high-load conditions, the motor's temperature escalates rapidly. Excessively high operating temperatures can disrupt the rate of chemical reactions and voltage output in fuel cells, leading to diminished efficiency or unreliable operation due to their sensitivity to temperature fluctuations [7]. This will impact the vehicle's total power performance. Consequently, the thermal dissipation issue of HPDPMSM is examined, highlighting the necessity for an effective cooling system design.

Extensive research has been conducted on the motor's cooling system to guarantee its dependable operation. The water cooling of the housing can efficiently lower the temperature of the stator winding. Researchers have developed multiple series of waterways for the waterway structure, which has significantly enhanced motor heat dissipation and optimized parameters including the number of waterways and the cooling water flow rate [8–10]. A unique water-cooling arrangement using cooling tubes within the stator yoke has been proposed to enhance thermal management. This design attained a maximum rotor temperature reduction of 11.17%, and the influence of coolant type and flow rate on heat exchange efficiency was comprehensively examined [11]. In comparison to the water-cooled frame under rated operating conditions, the integration of numerous water-cooled plates into the stator core layer can decrease the winding temperature by more than 20K [12]. The water-cooled frame efficiently lowers the thermal level of the stator winding, yet, the increased distance between the rotor and the frame leads to heightened thermal resistance. As a result, the temperature of the permanent magnet remains high.

A cooling structure utilizing a heat pipe was proposed to enhance the efficiency of temperature reduction in the machine. Three-dimensional heat pipes were inserted between the winding and the sleeve, and the heat pipes were secured with silicone gelatin. Augmenting the heat dissipation surface area to improve the cooling efficiency of the machine [13]. Wang, H., et al designed annular heat pipes inside the motor to cool its stator and rotor respectively [14]. The literature introduced a novel cooling system that integratively merges an airflow combination with a spiral channel design [15]. The coolant air may promptly disperse heat from the rotor, consequently enhancing temperature reduction efficiency. Chen, W., et al. examined the structure and environmental influences of the PMSM employed as the propulsion system for underwater UAVs, proposing a dual-channel water-cooled configuration integrated into the motor housing that obviates the necessity for auxiliary devices and promotes internal circulation for efficient heat dissipation within the motor. This method decreased the temperature of the permanent magnet by 12K [16]. A double-cycle arrangement utilizing external liquid cooling and internal air cooling has been developed to address the problem of internal heat dissipation within the motor. A cooling fan was affixed to the slotted stator yoke and spinning shaft to enhance internal airflow and mitigate temperature elevation. The permanent magnet's maximum temperature decreased by 13.5K [17]. Li, Y., et al. developed a motor cooling system

comprising an internal oil circulation and an external water circulation, with the latter utilizing a spoiler configuration. This method lowers the winding temperature by 5K compared to water cooling, hence markedly improving the motor's heat dissipation efficiency [18]. The complicated boundary conditions in the motor temperature field simulation significantly influence the findings. The Transformer model investigates the integration of simulation techniques for data and physics to improve the efficacy of temperature field simulation. This approach has substantially enhanced simulation precision and computational efficiency [19].

The article introduces a novel dual-waterway cooling system for the casing and rotor to address the heat dissipation issue in HPDPMSMs. The waterway's structural parameters are established by thermodynamic calculations, and a simulation model of the fluid-solid coupled temperature field is created. The efficacy of the dual-waterway cooling structure is validated by comparison with the cooling impact of a single waterway configuration. Simultaneously, the temperature distributions of the dual-waterway cooling structure are examined during various operating conditions, coolants, and water speeds. The designed frame-rotor dual-waterway cooling system is validated for its rationality.

## Materials and methods

### Cooling system design

A HPDPMSM with a rated power of 90kW, a rated speed of 6500 rpm, 8 poles, and 48 slots is selected as the subject of study. The primary parameters are presented in Table 1. The cooling system is analyzed. In order to limit the magnetic field disturbance and vibration noise, the double-layer winding structure is chosen. Increasing the machine's power density reduces the harmonic content of the magnetic field and enhances the efficient operational range of the machine.

The rotor employs a "V–" magnetic pole configuration to enhance the salient pole rate, thereby increasing the motor's reluctance torque, which subsequently improves torque density and efficiency. The permanent magnet material adopts the N48UH, which provides a high magnetic field strength for the motor. Table 2 presents the performance parameters of permanent magnets. Fig 1 depicts the configuration of the HPDPMSM.

The limited installation area for PMSMs in EVs, coupled with the demand for high power density designs, results in increased internal unit loss of the motor. The significant increase in temperature, combined with difficult heat dissipation in the operational environment, results in a persistent elevation of the motor's internal temperature. The conventional waterway structure is often constructed within the interior of the frame. The stator winding and rotor disperse heat through conduction. Relying exclusively on the single-waterway configuration of the frame is inadequate for efficiently reducing

**Table 1. Main parameters of HPDPMSM.**

| Parameter | Value |
| --- | --- |
| Rated power (kW) | 90 |
| Rated speed (r/min) | 6500 |
| Maximum speed (r/min) | 16000 |
| Number of poles | 8 |
| Number of slots | 48 |
| Outer diameter of stator (mm) | 220 |
| Inner diameter of stator (mm) | 143 |
| Outer diameter of rotor (mm) | 140.4 |
| Inner diameter of rotor (mm) | 84 |
| Air gap length (mm) | 1.3 |
| Core length (mm) | 126 |
| Outer diameter of rotor support cylinder (mm) | 84 |
| Inner diameter of rotor support cylinder (mm) | 50 |

**Table 2. Performance parameters of permanent magnets.**

| Parameter | Maximum magnetic energy product (MGOe) | Remanence (kGs) | Magnetic coercivity (KOe) | Intrinsic coercivity (kOe) | Reversible Temperature Coefficient (%/°C) | | Max Operating Temperature (°C) |
|---|---|---|---|---|---|---|---|
| | | | | | \|αBr\| | \|βHcj\| | |
| N48UH | 47 | 13.8 | 13.3 | 25 | 0.11 | 0.5 | 180 |

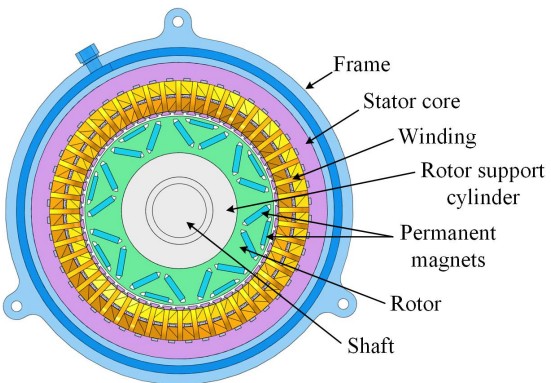

**Fig 1. Structure of HPDPMSM.**

rotor temperature escalation. This constraint can undermine the dependability of motor function and may lead to irreversible demagnetization of the permanent magnet.

A frame-rotor dual-waterway cooling structure is proposed to enhance the cooling capacity of the HPDPMSM cooling system. The outer canal constitutes the spiral waterway of the frame, whereas the inner waterway is a spiral waterway positioned within the rotor support cylinder. Fig 2 depicts the cooling system. The contact surface area between the coolant and the rotor is significantly increased, improving thermal conductivity and efficiently lowering the temperature rise in different sections of the rotor.

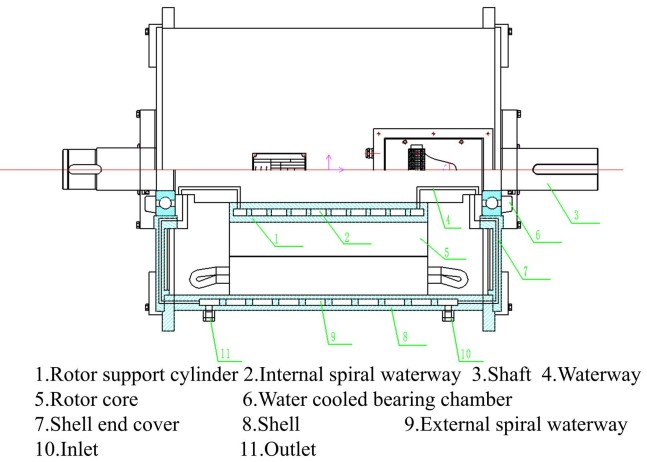

1.Rotor support cylinder 2.Internal spiral waterway 3.Shaft 4.Waterway
5.Rotor core 6.Water cooled bearing chamber
7.Shell end cover 8.Shell 9.External spiral waterway
10.Inlet 11.Outlet

**Fig 2. Frame-rotor dual-waterway cooling system.**

The outer waterway consists of a inlet, a spiral water channel, and a outlet within the frame. The coolant circulates within to absorb and dissipate the heat produced by the stator iron losses and the winding copper losses via heat conduction. The increase in stator temperature is significantly mitigated. The coolant traverses the end cap, bearing chamber, shaft, and rotor waterway within the inner waterway. The interface area between the coolant and the rotor is markedly enhanced. The thermal conductivity is enhanced, resulting in a significant reduction in the temperature rise of each rotor component.

The spiral channel is designated for both the outer and inner waterways. The coolant's passage through the spiral channel generates a pressure differential, facilitating continuous circulation and enhancing heat dissipation efficiency. The coolant circulates around the machine, effectively enveloping the stator and rotor components, so ensuring rapid and uniform heat dissipation. The intricate design of the internal canal has been simplified to a spiral configuration, with inlets and outlets on either side. Fig 3 illustrates the cooling system.

The original waterway configuration is established using the thermodynamic analysis of the waterway [20]. The design of the waterway must account for both fluid flow characteristics and heat exchange efficiency [21]. The heat transfer coefficient and flow resistance are as follows.

$$Q = \frac{P_i}{\rho C_w \Delta t}$$

(1)

Where, $Q$ is the flow rate of cooling water, $P_i$ represents the total loss, excluding mechanical loss, $\rho$ is the density of liquid, $C_w$ is the specific heat capacity of water, $\Delta t$ is the temperature difference between inlet and outlet.

The formula for calculating the Reynolds coefficient.

$$Re = \frac{Q D_e}{A \nu_w}$$

(2)

Where, $D_e$ is the equivalent diameters at different positions of the waterway, $A$ is cross-sectional area of the waterway, $\nu_w$ is the flow rate of cooling water in the waterway.

Waterway cross-sectional area:

$$A = ab$$

(3)

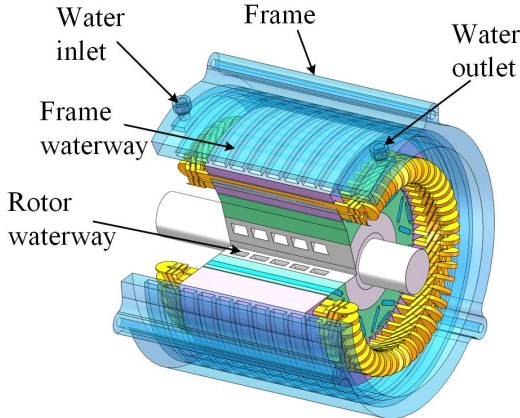

**Fig 3. Frame-rotor dual-waterway cooling simplified system.**

Where, $a$ is the channel width, $b$ is the channel height.

The circumference of the waterway section

$$p = 2(a + b) \tag{4}$$

Water velocity:

$$v_w = \frac{Q}{A} \tag{5}$$

Equivalent diameter

$$d_e = 4\frac{A}{p} \tag{6}$$

The fluid in the channel, propelled by the pump, is in a turbulent condition. The channel heat transfer coefficient is defined as the forced convection heat dissipation coefficient between the channel wall and the cooling water.

$$\alpha_j = \frac{Nu_f \lambda_w}{D_e} \tag{7}$$

Where, $Nu_f$ is the Nusselt number of the fluid, $\lambda_w$ is the thermal conductivity of water,

Heat absorbed by cooling water:

$$\Phi = \alpha_j S \Delta t_j \tag{8}$$

Where, $S$ is the cooling wall area, $\Delta t_j$ is the temperature difference between water and the cooling wall.

Inlet and outlet pressure difference:

$$\Delta p = \rho g h + \rho g (h_f + h_w) \tag{9}$$

Where, $g$ is the acceleration of gravity, $h$ is the height difference of inlet and outlet, $h_f$ is the resistance along the path, $h_w$ is the local resistance.

$$h_f = \lambda \frac{L_{all}}{d_e} \frac{v_w{}^2}{2g} \tag{10}$$

Where, $\lambda$ is resistance coefficient, $L_{all}$ is total length of waterways.

Total length of waterways

$$L_{all} = \frac{\pi D L}{a + c} \tag{11}$$

Where, $D$ is the diameter of the circle where the center of the waterway is located, $L$ is the motor axial length, $c$ is the interval length between adjacent waterways.

When $2300 < R_e < 10^5$, resistance coefficient $\lambda$:

$$\lambda = \frac{0.3164}{R_e^{0.25}}$$

<div align="right">(12)</div>

When $10^5 < R_e < 3 \times 10^6$:

$$\lambda = 0.0032 + \frac{0.221}{R_e^{0.237}}$$

<div align="right">(13)</div>

The waterways in the motor usually meet: $2300 < R_e < 10^5$, so $\lambda = \frac{0.3164}{R_e^{0.25}}$
The waterway structure is essential for the thermal regulation of motors. By refining the design, the coolant may be uniformly channeled through the motor's essential components, thereby efficiently dispersing the heat produced during operation. Considering these elements, accurate computations enable the assessment of multiple parameters of the dual-waterway structure, as depicted in Fig 4. The external water channel is 13 mm in breadth, 6 mm in height, and consists of 11 channels. The inner water channel has a width of 15 mm, a height of 8 mm, and consists of 5 channels.

## Mathematical model of temperature field

### Heat source calculation

During electromechanical energy conversion in the motor, various losses primarily manifest as heat, resulting in a temperature increase. The losses of PMSM generated by EV mostly include winding copper loss, stator core loss, permanent magnet eddy current loss, mechanical loss, and stray loss. The little harmonic content of the winding magnetomotive force allows for the disregard of rotor iron loss and stray loss [22].

$$P_{cu} = mI^2 R$$

<div align="right">(14)</div>

Where, $P_{cu}$ is the winding copper loss, $m$ is the phase number, $I$ is the effective value of the motor phase current, $R$ is the winding resistance.

$$P_{Fe} = C_h f B_m^n + C_e f^2 B_m^2 + C_{ex} f^{1.5} B_m^{1.5}$$

<div align="right">(15)</div>

Where, $P_{Fe}$ is the stator iron loss; $C_h$ is the hysteresis loss coefficient; $C_e$ is the eddy current loss coefficient, $C_{ex}$ is the additional loss coefficient, $f$ is magnetic field frequency, $B_m$ is the maximum magnetic density.

$$P_{pm} = \int_V \frac{J^2}{\sigma} dV$$

<div align="right">(16)</div>

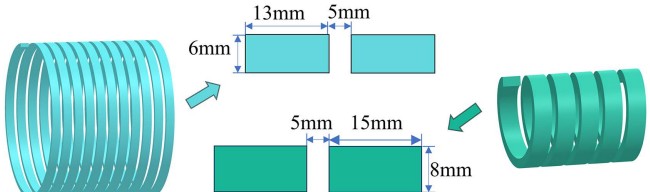

**Fig 4. Dual-waterway structure.**

Where, $P_{pm}$ is the permanent magnet eddy current loss, $J$ is current density, $\sigma$ is the electrical conductivity, $V$ is the region of spatial integration.

$$P_f = 0.15 \frac{F}{d} \nu_f \times 10^{-5}$$

(17)

Where, $P_f$ is mechanical loss of bearing, $F$ is bearing load, $d$ is the diameter of the shaft, $\nu_f$ is the circular velocity at the center of the shaft.

The mechanical loss is influenced by the motor's speed and is predominantly concentrated in the bearing. The water-cooled design of the bearing chamber can significantly mitigate the effects of mechanical loss. The losses and heat generation of several components in the 90kW HPDPMSM at rated operating circumstances are specified in Table 3. The fluid-solid coupled temperature field of the HPDPMSM is examined based on the heat productivity calculation results and the planned dual-waterway structure.

**Heat transfer model**

The methods of heat transfer in the operation of PMSM are divided into conduction, radiation, and convection. The differential equation for three-dimensional steady-state heat conduction in the motor is as follows [23]:

$$\begin{cases} \frac{\partial}{\partial x}(\lambda_x \frac{\partial T}{\partial x}) + \frac{\partial}{\partial y}(\lambda_y \frac{\partial T}{\partial y}) + \frac{\partial}{\partial z}(\lambda_z \frac{\partial T}{\partial z}) = -q_v \\ \frac{\partial T}{\partial n}\big|_{S_1} = 0 \\ -\lambda_m \frac{\partial T}{\partial n}\big|_{S_2} = \alpha(T - T_f) \end{cases}$$

(18)

Where, $\lambda_x$, $\lambda_y$ and $\lambda_z$ are thermal conductivity of various media in the motor in x, y, z directions, $\lambda_m$ is thermal conductivity of material, $q_v$ is the heat source density, $\alpha$ is the convective heat dissipation coefficient, $T_f$ is the ambient fluid temperature, $T$ is the temperature, $S_1$ and $S_2$ are object boundaries.

According to the theory of heat transfer, the flow of coolant in the motor water cooling system satisfies the law of mass conservation, energy conservation and momentum conservation.

Mass conservation:

$$\frac{\partial(\rho u)}{\partial x} + \frac{\partial(\rho v)}{\partial y} + \frac{\partial(\rho w)}{\partial z} = 0$$

(19)

Where, $u$, $v$ and $w$ are velocity components.

Energy conservation:

$$\frac{\partial(\rho T)}{\partial t} + div(\rho u T) = div(\frac{k}{c_p} grad T) + S_t$$

(20)

**Table 3. The loss and heat productivity of each part of the motor under rated working condition.**

| Motor component | Loss (W) | Effective volume (m³) | Heat productivity (W/m³) |
|---|---|---|---|
| Stator | 1153.57 | 0.00202 | 571074.25 |
| Permanent magnet | 63.43 | 0.00024 | 264291.67 |
| Winding | 1806.70 | 0.00041 | 4406585.37 |

Where, $k$ is the fluid heat transfer coefficient, $S_t$ represents the viscous dissipation term, $c_p$ is the specific heat capacity.

Momentum conservation:

$$\begin{cases} \frac{\partial(\rho u)}{\partial t} + \frac{\partial(\rho uu)}{\partial x} + \frac{\partial(\rho vu)}{\partial y} + \frac{\partial(\rho wu)}{\partial z} = \\ -\frac{\partial p}{\partial x} + \frac{\partial \tau_{xx}}{\partial x} + \frac{\partial \tau_{yx}}{\partial y} + \frac{\partial \tau_{zx}}{\partial z} + F_x \\ \frac{\partial(\rho v)}{\partial t} + \frac{\partial(\rho uv)}{\partial x} + \frac{\partial(\rho vv)}{\partial y} + \frac{\partial(\rho wv)}{\partial z} = \\ -\frac{\partial p}{\partial y} + \frac{\partial \tau_{xy}}{\partial x} + \frac{\partial \tau_{yy}}{\partial y} + \frac{\partial \tau_{zy}}{\partial z} + F_y \\ \frac{\partial(\rho w)}{\partial t} + \frac{\partial(\rho uw)}{\partial x} + \frac{\partial(\rho vw)}{\partial y} + \frac{\partial(\rho ww)}{\partial z} = \\ -\frac{\partial p}{\partial z} + \frac{\partial \tau_{xz}}{\partial x} + \frac{\partial \tau_{yz}}{\partial y} + \frac{\partial \tau_{zz}}{\partial z} + F_z \end{cases} \tag{21}$$

Where, $p$ is the pressure on the fluid element, $\tau_{xy}, \tau_{yy}$ and $\tau_{zy}$ are the components of the viscous stress exerted on the surface of the microelement due to the viscous action of the molecule, $F_x, F_y$ and $F_z$ are volume forces on the microelement.

The heat conduction and heat convection of the designed motor are the main research objects and the influence of heat radiation is ignored. The $k$ -$\varepsilon$ model is used for simulation calculation, and the equation is as follows:

$$\begin{cases} \frac{\partial(\rho k)}{\partial t} + \frac{\partial(\rho k u_i)}{\partial x_i} = \frac{\partial}{\partial x_j}[(\mu + \frac{\mu_i}{\alpha_k})\frac{\partial k}{x_j}] + G_k + G_b - \rho\varepsilon - Y_m + S_k \\ \frac{\partial(\rho \varepsilon)}{\partial t} + \frac{\partial(\rho \varepsilon u_i)}{\partial x_i} = \frac{\partial}{\partial x_j}[(\mu + \frac{\mu_i}{\alpha_\varepsilon})\frac{\partial \varepsilon}{x_j}] + G_{1\varepsilon}\frac{\varepsilon}{k}(G_k + G_{3\varepsilon}G_b) - G_{2\varepsilon}\rho\frac{\varepsilon^2}{k} + S_\varepsilon \end{cases} \tag{22}$$

Where, $G_k$ is the turbulence kinetic energy generated by laminar velocity gradient, $G_b$ is the turbulence kinetic energy generated by buoyancy, $Y_m$ is the fluctuation caused by excessive diffusion in compressible turbulence, $G_{1\varepsilon}$, $G_{2\varepsilon}$ and $G_{3\varepsilon}$ are the empirical coefficients, $\alpha_k$ and $\alpha_\varepsilon$ are the turbulent Prandtl numbers of turbulence kinetic energy $k$ equation and dissipation rate $\varepsilon$ equation respectively, $S_k$ and $S_\varepsilon$ are custom items.

## Basic assumptions

The HPDPMSM is modeled in conjunction with the cooling structure parameters derived from the water thermodynamic calculation. To streamline the subdivision and simulation computation process, the subsequent assumptions are established.

1. The heat losses and temperature rise of the motor components are evenly distributed according to the body.

2. The insulation layer in the stator slots and the winding insulation are equivalent to one insulation layer and evenly distributed in the slots.

3. The multi-turn winding in the double-layer winding is equivalent to two complete copper conductors, and the end of the winding is replaced by the extended lengths.

4. The end cover and bearing chamber of the inner waterway structure are simplified to lead out the water inlet and outlet at both end of the rotor support cylinder.

## Calculation of thermal conductivity

In order to improve the simulation accuracy, the air thermal conductivity inside the motor needs to be calculated.

$$R_{ecr} = 41.2\sqrt{(R_i/\delta)} \tag{23}$$

Where, $R_i$ is stator inner diameter, $\delta$ is air gap length.

When $R_e < R_{ecr}$, the air flow in the air gap is laminar, the effective thermal conductivity is equal to the thermal conductivity of the air.

When $R_e < R_{ecr}$, the air flow in the air gap is turbulent, and the formula for calculating the effective thermal conductivity is formula (24).

$$\lambda_{eff} = 0.0019\eta^{-2.9084}R_e^{\,0.4614\ln(3.33361\eta)}$$

(24)

The core is typically constructed from thin silicon steel sheets laminated along the axial direction, resulting in pronounced thermal conductivity anisotropy. In the axial direction, the heat conduction process can be modeled as series heat transfer between multiple planar walls. In contrast, in the circumferential and radial directions, it can be approximated as parallel heat transfer between multiple planar walls. The anisotropic thermal conductivity of the core is expressed as follows.

$$\lambda_{aT} = (\delta_{Fe} + \delta_i)\Big/\Big(\frac{\delta_{Fe}}{\lambda_{Fe}} + \frac{\delta_i}{\lambda_i}\Big) = 1\Big/\Big(\frac{K_{Fe}}{\lambda_{Fe}} + \frac{1-K_{Fe}}{\lambda_i}\Big)$$

(25)

$$\lambda_{rT} = K_{Fe}\lambda_{Fe} + (1-K_{Fe})\lambda_i$$

(26)

Where, $\lambda_{aT}$ is axial thermal conductivity of the iron core, $\lambda_{rT}$ is the circumferential and radial thermal conductivity of the core, $\delta_{Fe}$ is thickness of laminated core, $\delta_i$ is thickness of insulating medium, $K_{Fe}$ is the lamination coefficient of the core, $\lambda_i$ is the thermal conductivity of insulating media, $\lambda_{Fe}$ is the thermal conductivity of the laminated core.

Calculation of the equivalent thermal conductivity of slot insulation.

$$\lambda_{eq} = \frac{\sum\limits_{i=1}^{n} d_i}{\sum\limits_{i=1}^{n} \frac{d_i}{\lambda_i}}$$

(27)

Where, $\lambda_{eq}$ is the thermal conductivity of equivalent insulation, $d_i$ is the equivalent thickness of each insulating material in the slot, $\lambda_i$ is the thermal conductivity of insulating materials.

The HPDPMSM physical model of the fluid-solid coupled temperature field is constructed, as illustrated in Fig 5.

A B

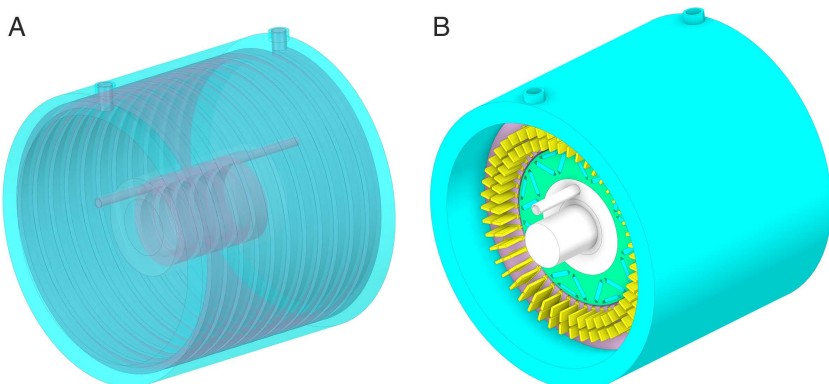

**Fig 5. Temperature field solution domain model of HPDPMSM. (A) Dual-waterway structure, (B) Motor 3D model.**

## Result

### Simulation of fluid-solid coupled temperature field

Ansys Fluent software is employed to simulate the machine model [24–25]. The dual-waterway structure is isolated from all machine components, and water is utilized as the coolant for simulation analysis because to its high thermal conductivity.

The simulation study utilizes water as the coolant, with the ambient temperature established at 300K. Considering the constant flow of cooling water during the motor's operation, the inlet temperature is established at 333K. The flow rate is established at 0.5m/s, with the outlet defined as a free outlet boundary condition.

### Temperature distributions with different waterway structures

To verify the high efficiency of the engineered cooling system, temperature field simulations are performed separately for the dual-waterway structure and the single-waterway structure. The solitary canal structure solely contains the water within the confines of the frame, while other characteristics remain unaltered.

Fig 6 illustrates the temperature distributions of the winding for the two canal structures in the HPDPMSM at the rated speed of 6500r/min. The contact between the winding end and the air within the machine cavity results in significant thermal resistance between the winding and the water-cooled structure. The winding end complicates heat dissipation and temperature regulation. The dual-waterway configuration can sustain a more stable electromagnetic performance in high-temperature conditions.

Fig 6A illustrates the winding temperature distribution with a single waterway, where the maximum temperature reaches 399.9K. Fig 6B illustrates the temperature distribution of the winding employing the dual-waterway system, indicating a maximum temperature of 394K. In comparison to the single waterway, its temperature rise is diminished by 5.9K.

Fig 7 depicts the temperature distributions of the stator in the HPDPMSM with dual waterway. The temperature of the stator teeth is higher due to the heat generated by the winding. The stator yoke is tightly affixed to the frame and is directly cooled by water, resulting in a low temperature. The maximum stator temperature for the single waterway configuration is 375.2K, whereas for the dual-waterway structure, it is 369.8K, indicating a temperature rise reduction of 5.4K.

Fig 8 illustrates the temperature distributions of the rotor for the two canal layouts. The maximum temperatures observed for the rotor core and permanent magnet in a single canal design attain 392.3K. The principal thermal source

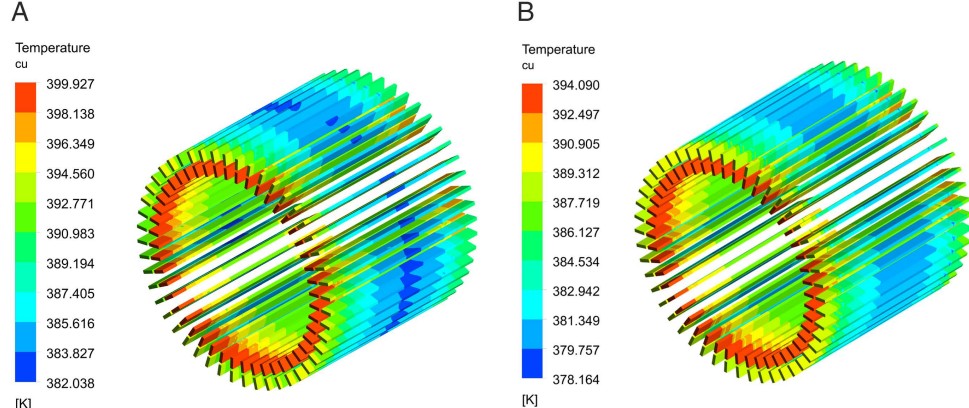

**Fig 6. Winding temperature distributions. (A) Single waterway winding, (B) Dual-waterway winding.**

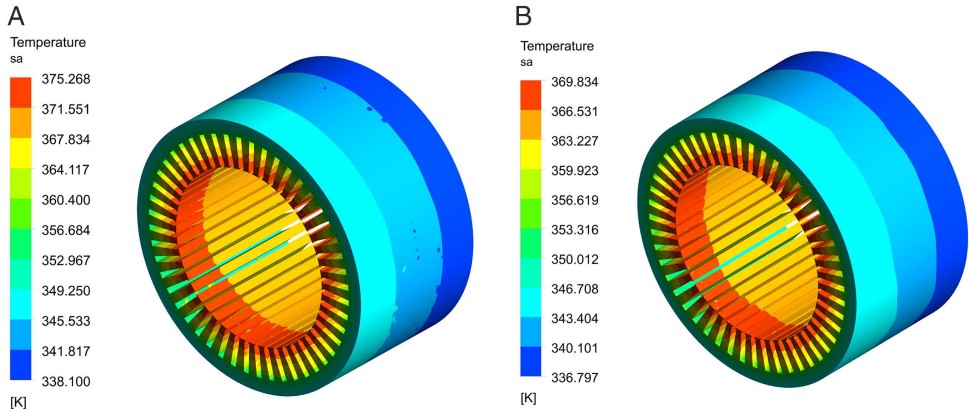

**Fig 7. Stator temperature distributions.** (A) Single waterway stator, (B) Dual-waterway stator.

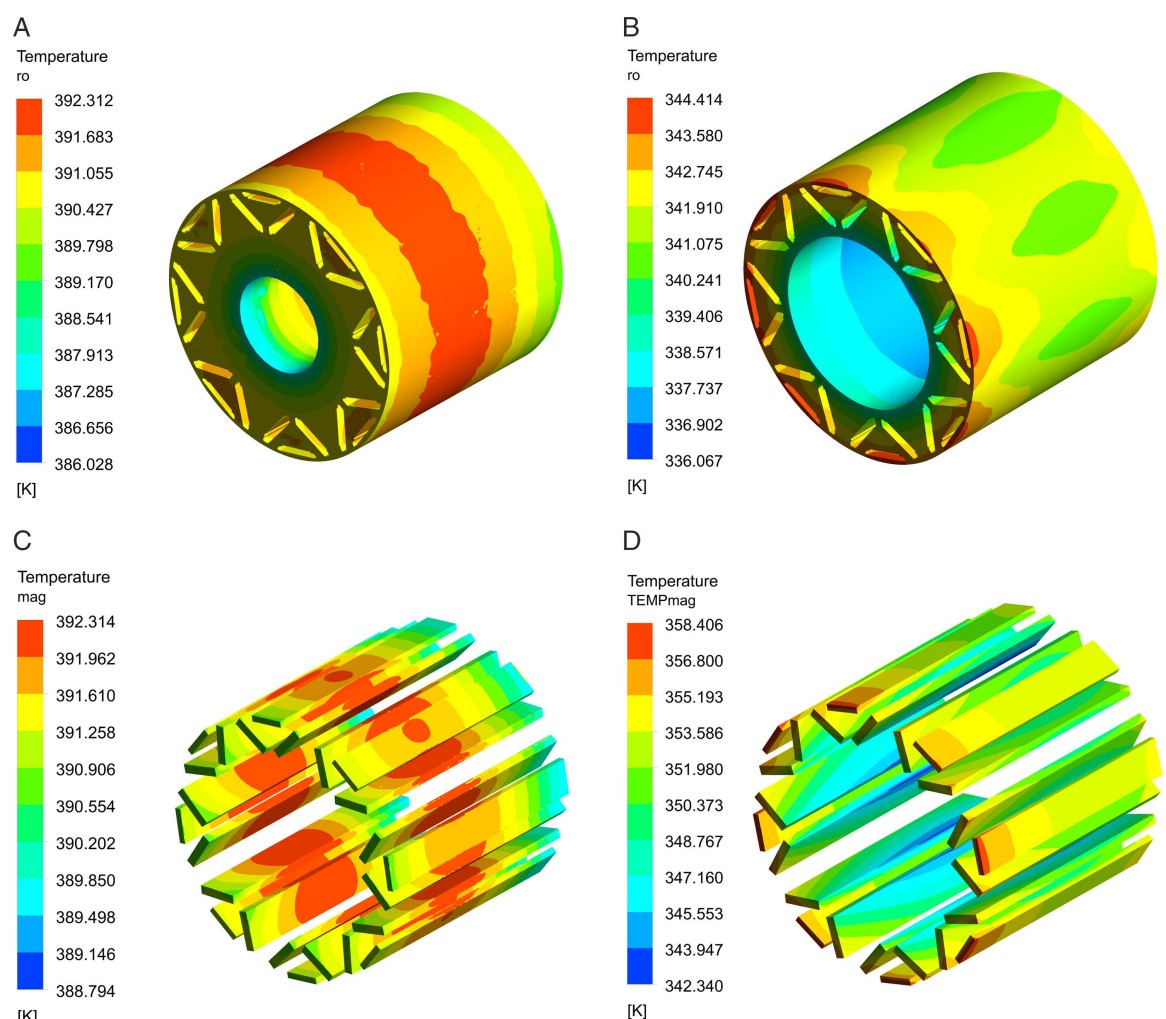

**Fig 8. Rotor temperature distributions.** (A) Single waterway rotor core, (B) Dual-waterway rotor core, (C) Single waterway permanent magnet, (D) Dual-waterway permanent magnet.

of the rotor is the eddy current loss produced by the permanent magnet, intensified by the difficult heat dissipation in the middle area. As a result, the central region of the permanent magnet displays the maximum temperature. In the dual-waterway structure, the maximum temperatures of the rotor core and permanent magnet reach 344.4K. The rotor temperature rise is diminished by 47.9K, while the heat dissipation efficiency is enhanced by 12%.

The results satisfy the operational temperature criteria for N48UH permanent magnet material. The dual-waterway structural design facilitates increased coolant flow within the rotor support cylinder, hence improving the efficiency of heat dissipation generated by the permanent magnet. The temperature gradient within the rotor demonstrates a progressive decrease from the outer surface to the inner surface, due to the cooling influence of the inner canal. Fig 9 depicts the water temperature distributions for the two designs of the waterway. The temperature differential between the inlet and outlet in the single waterway is documented at 13.22K, while in the dual waterway, this differential is 11.94K, accompanied by an additional temperature difference of 5K noted in the inner waterway.

In comparison to the heat generated by the stator core and the winding, the heat produced by the permanent magnet is little. The temperature differential in the inner canal is inferior to that of the outer. Moreover, with the inlet flow rate set at 0.5m/s, the overall thermal management efficacy of the dual-waterway configuration surpasses that of the single-waterway system.

Fig 10 depicts the water velocities in the two waterway layouts, whereas Fig 11 demonstrates the associated water pressure distributions. At the identical inlet flow rate, the dual-waterway structure exhibits the highest water velocity and pressure, with maximum values of 1.23m/s and 16.5kPa, respectively. Its pressure exceeds that of the single canal structure by 5.3kPa, yet, it complies with the specifications of the waterway structure.

## Experimental facility

Fig 12 shows the motor test platform. The controller transforms direct current into alternating current to provide power for the motor. The cooling duct delivers coolant to the motor, while the induction motor imposes a load on the test motor to verify that it functions under rated conditions.

The test environment temperature is 28°C, and the simulated environment temperature is set at 300K. Fig 13 shows the relevant results. Fig 13A shows the maximum temperature variation of the motor winding in the motor experiment.

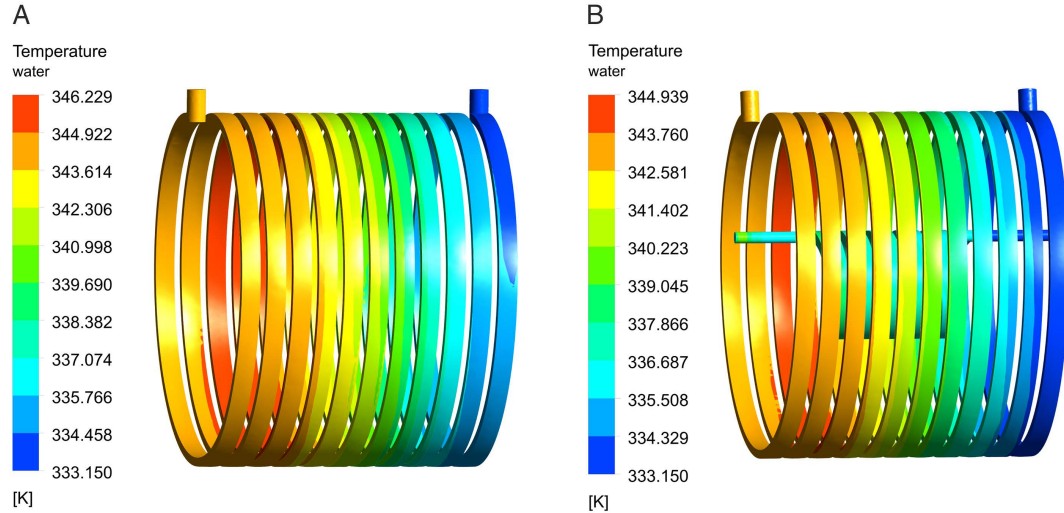

**Fig 9. Temperature distributions of waterways. (A) Single waterway, (B) Dual-waterway.**

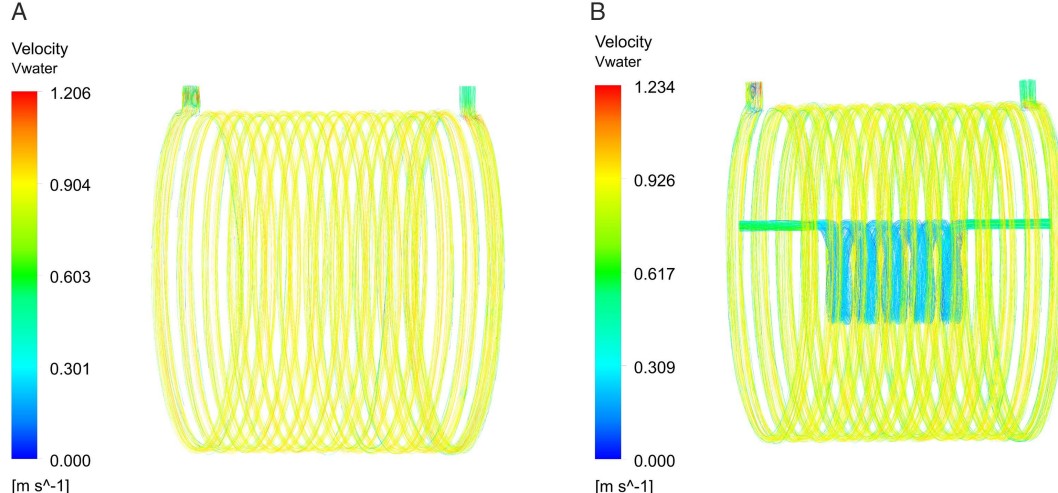

**Fig 10. Distributions of water velocities.** (A) Single waterway velocity distribution, (B) Dual-waterway velocity distribution.

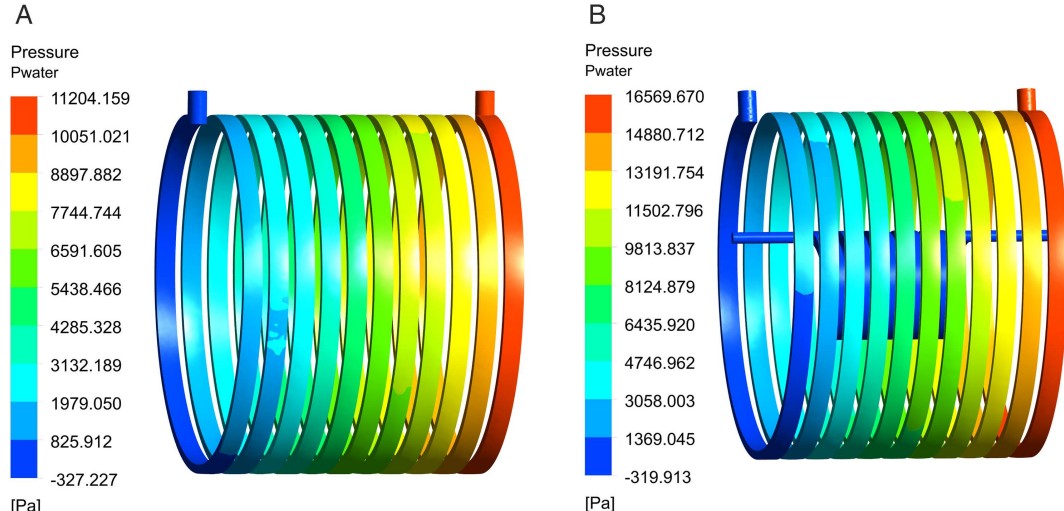

**Fig 11. Distributions of water pressure.** (A) Single waterway pressure distribution, (B) Dual-waterway pressure distribution.

After stabilization, the maximum temperature of the winding is 148.63°C, with a temperature rise of 120°C. Fig 13B shows the simulation results of the motor temperature field. It has been observed that a 5.76K temperature rise discrepancy exists between the test and simulation results, primarily due to the simplification of the motor winding's end during the simulation process. However, the relative error of their results being less than 5% proves the validity of the simulation model, which further proves the rationality of the simulation model of the dual waterway.

## Temperature distributions under different coolants

Liquid coolants generally possess a thermal conductivity that is considerably superior to that of air, hence markedly improving the system's heat absorption capability during the heat exchange process. Moreover, during flow, the cooling medium must comply with the principle of conservation of energy. As indicated in Eq (20), fluctuations in the physical

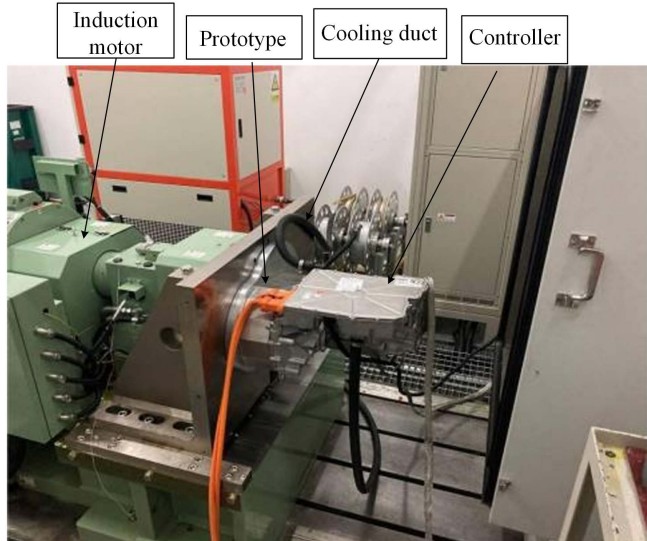

**Fig 12. Motor test platform.**

characteristics of the cooling medium—namely density, specific heat capacity, and viscosity—directly affect its heat exchange efficiency, consequently influencing the efficacy of its heat dissipation performance.

The choice of coolant substantially influences the motor's cooling efficiency. Four prevalent coolants for automotive engines—water, 30% ethylene glycol solution, 50% ethylene glycol solution, and oil—are chosen to model the temperature distribution of HPDPMSM during the rated functioning of the dual-water cooling system. The coolant specifications are presented in Table 4. Fig 14 illustrates the temperature distribution.

The maximum temperature observed for a motor utilizing water cooling is 394K. The maximum temperatures for the motor at different concentrations of ethylene glycol solution are 394.8K for a 30% solution and 395.2K for a 50% solution. Moreover, the maximum temperature during oil cooling attains 404.83K.

An analysis is conducted on the impact of different coolants on the maximum temperature rise of the stator, rotor, permanent magnet, and winding in relation to an ambient temperature of 300K, as illustrated in Fig 15. Among the four coolants—water, 30% ethylene glycol solution, 50% ethylene glycol solution, and cooling oil—the cooling efficacy diminishes in succession. The cooling effect is optimal due to water's superior thermal conductivity.

The highest temperature rise in the winding and permanent magnet when utilizing ethylene glycol solution as the coolant is only marginally more than that recorded with water. With an increase in the concentration of ethylene glycol solution, the winding temperature elevates by 0.5K, while the temperature of the permanent magnet ascends by 0.6K when utilizing a 50% ethylene glycol solution in comparison to a 30% solution. The heat conductivity of oil is the lowest, resulting in the least effective cooling. The dual-waterway structure segregates the coolant from the motor components, allowing for the selection of water cooling without regard to the coolant's insulation. In low-temperature regions, ethylene glycol solution can be employed in EVs to keep the cooling system from freezing.

## Temperature distributions for different velocities

Water velocity is a crucial determinant of the motor's heat dissipation efficiency, while water pressure is a vital metric for assessing the cooling system's efficacy. Consequently, the cooling effect and water pressure of the motor are examined by altering the inlet flow rate while keeping other parameters constant. The velocity range is established at 0.2 to 1.2m/s.

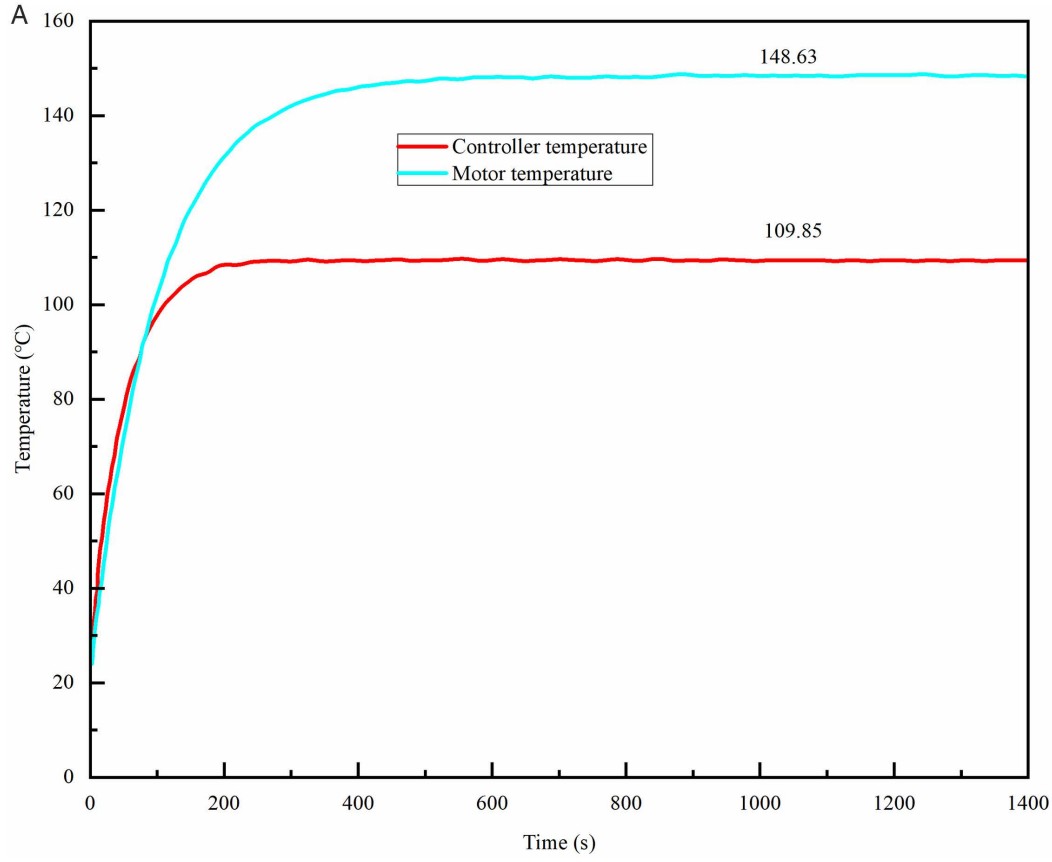

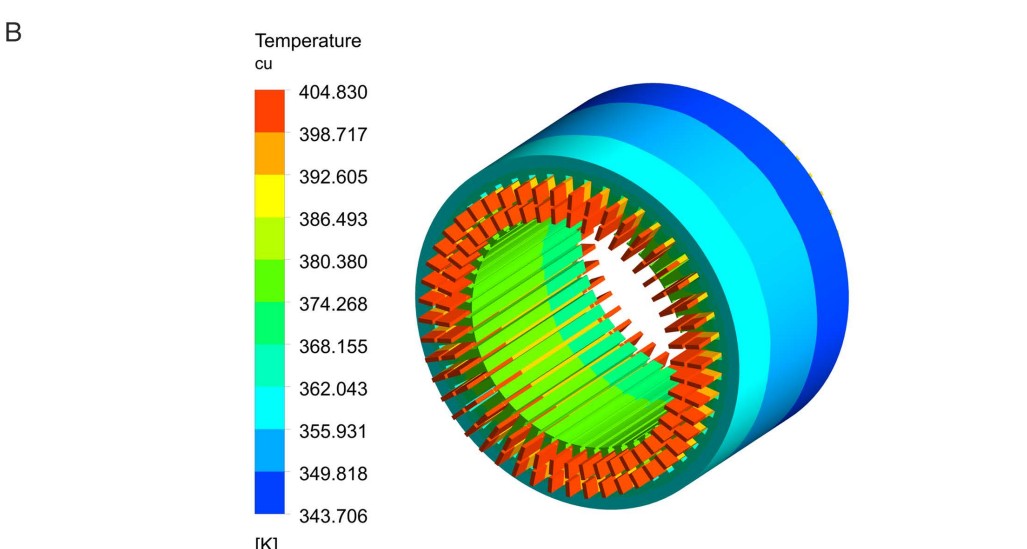

**Fig 13. Experiment and simulation comparison under rated working conditions. (A) Experiment result, (B) Simulation result.**

**Table 4. Parameters of coolants.**

| Coolant | Density (kg/m³) | Specific heat capacity (J/(kg·K)) | Thermal conductivity (W/(m·K)) | Viscosity coefficient (Kg/(m·s)) |
|---|---|---|---|---|
| Water | 998.2 | 4182 | 0.6 | 0.001003 |
| oil | 884 | 2320 | 0.1358 | 0.0014144 |
| 30% ethylene glycol solution | 1020.42 | 3787 | 0.494 | 0.00194 |
| 50% ethylene glycol solution | 1068.75 | 3474 | 0.408 | 0.00298 |

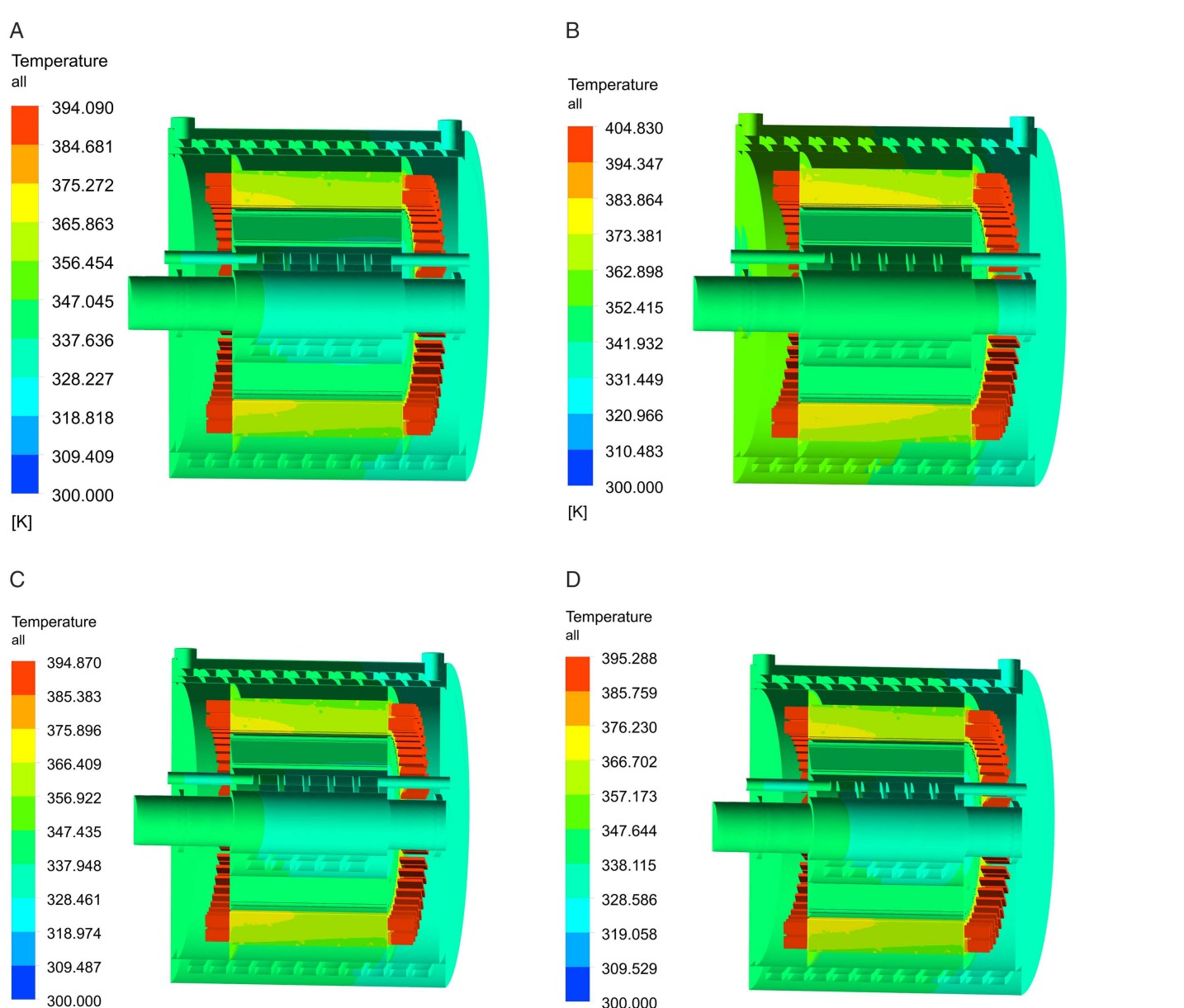

**Fig 14. Temperature distributions under different coolants.** (A) Water, (B) Cooling oil, (C) 30% ethylene glycol solution, (D) 50% ethylene glycol solution.

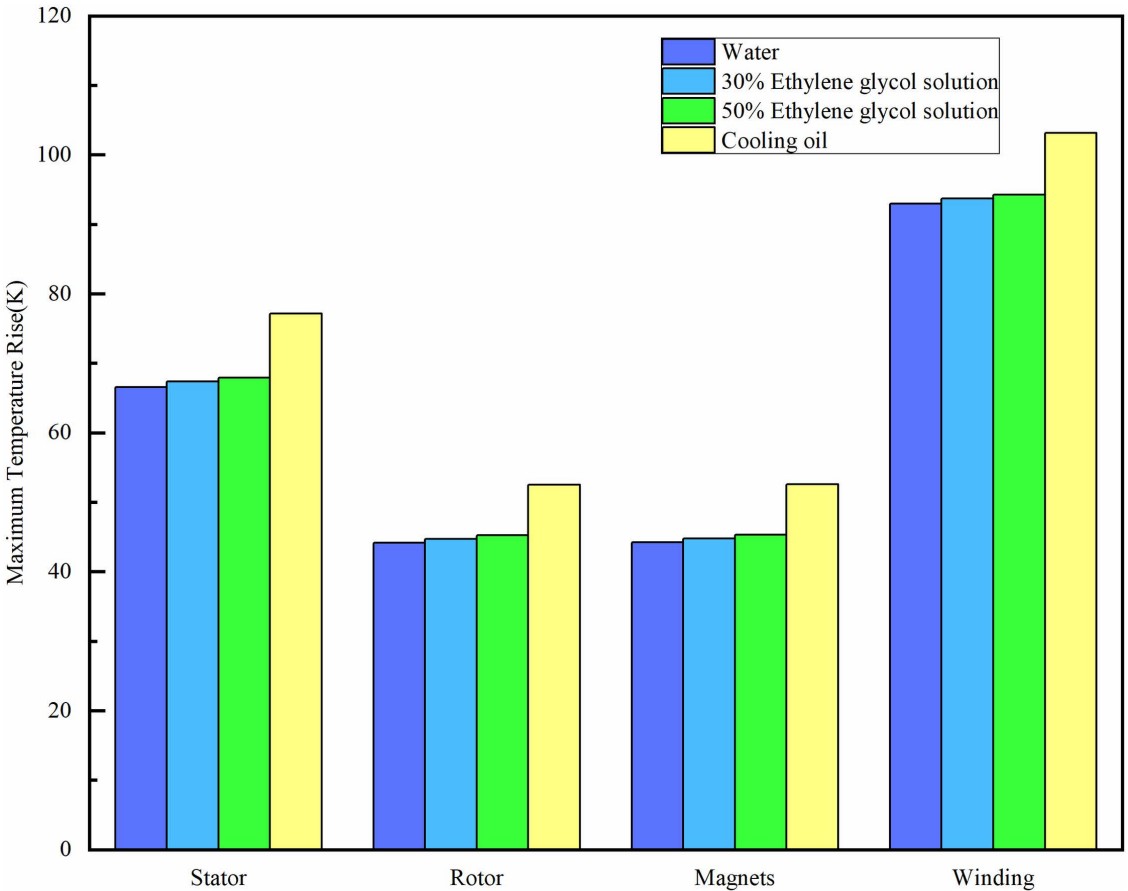

**Fig 15. Comparison of temperature rise of motor components under different coolants.**

The highest temperature rise of the winding and permanent magnet, and the highest-pressure distributions in the waterway for different velocities are obtained, as shown in Fig 16. To achieve an effective cooling effect while satisfying the strength requirements of the waterway structure, the optimal velocity for the dual-waterway configuration is determined to be 0.5m/s.

At low velocities, an increase can markedly reduce the temperature elevation. When the velocity reaches 0.5m/s, the declining trend of the maximum temperature rise of the winding and permanent magnet progressively diminishes. However, the maximum pressure within the waterway increases approximately linearly with the growth in velocity.

## Temperature distributions at different operating conditions

The operational efficacy and reliability of motors have consistently been critical factors in motor design and development. Particularly at high speeds, EVs experience an augmented electromagnetic strain on the motor, resulting in heightened iron and copper losses, which subsequently elevate the motor's temperature. To guarantee the dependable functioning of EVs within the specified speed range, thermal simulations of the motor are performed under conditions of maximum speed and maximum power operation.

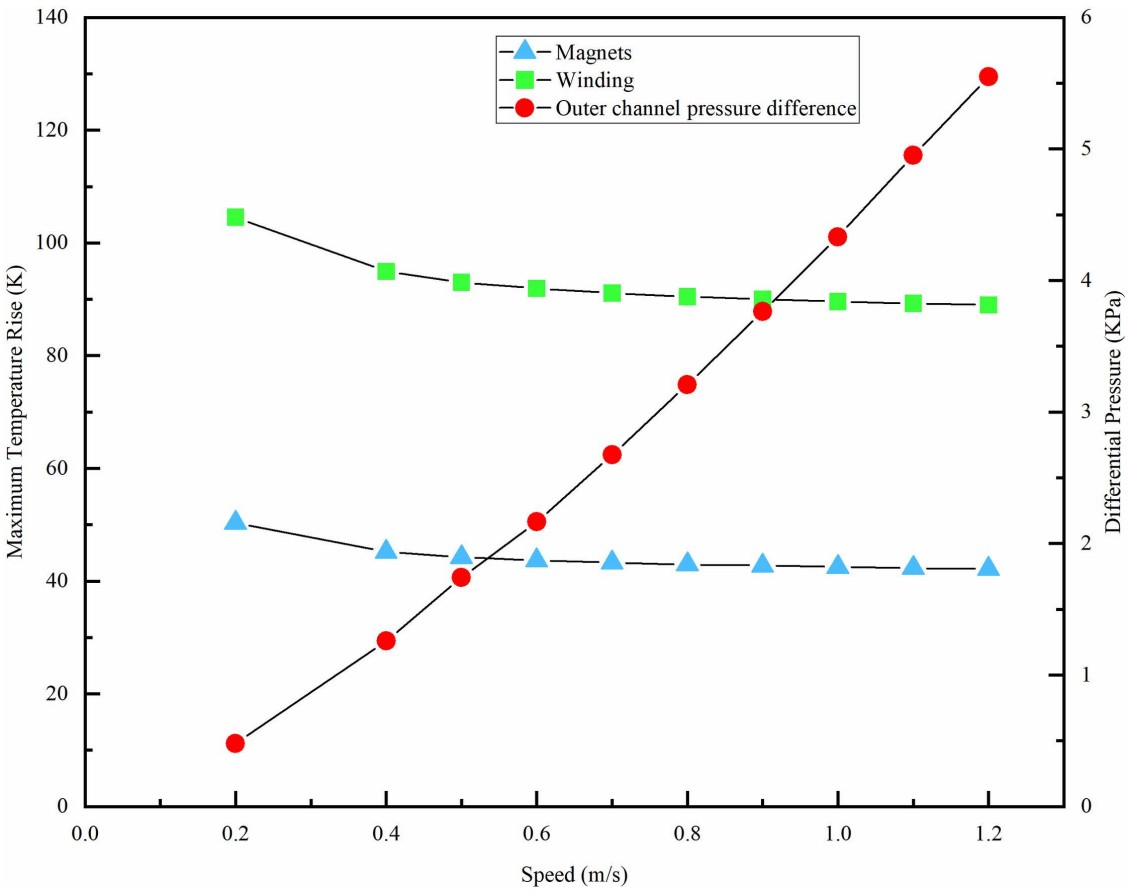

**Fig 16. Winding and permanent magnet temperature for different velocities.**

The instantaneous temperature field variations of the motor were studied under stable temperature conditions at rated operating conditions, with the rotational speed increased to the maximum of 16000 rpm and a peak power of 240kW. Under the condition of an ambient temperature of 300K, the temperature rise of the motor at its maximum speed is shown in Fig 17. The maximum temperature of the motor winding reaches 438K, showing a temperature rise of 138K, which is 45.23K higher than under rated operating conditions. The maximum temperature of the permanent magnet is 358K, with a temperature rise of 58K, 14.26K higher than the rated operating condition. Fig 18 illustrates the temperature rise of the motor at peak power of 240kW. At maximum power conditions, the motor winding reaches a peak temperature of 442K, with a temperature rise of 142K, while the permanent magnet attains a maximum temperature of 361K, with a temperature rise of 61K. This temperature rise change complies with the maximum operating temperature requirement for the N48UH permanent magnet material, and meets the temperature rise limits allowed for Class H insulation under short-term operating conditions.

The results indicate that, in extreme working conditions, the dual-waterway structure efficiently regulates the motor's internal temperature distribution, preventing overheating and guaranteeing stable performance and long-term reliability at high power and speed levels.

The aforementioned multi-condition transient simulation results of the motor can furnish data support for the motor dynamic temperature prediction algorithm, thereby facilitating a more comprehensive examination of the algorithm's practicality in assessing motor operating conditions and forecasting motor lifespan [26].

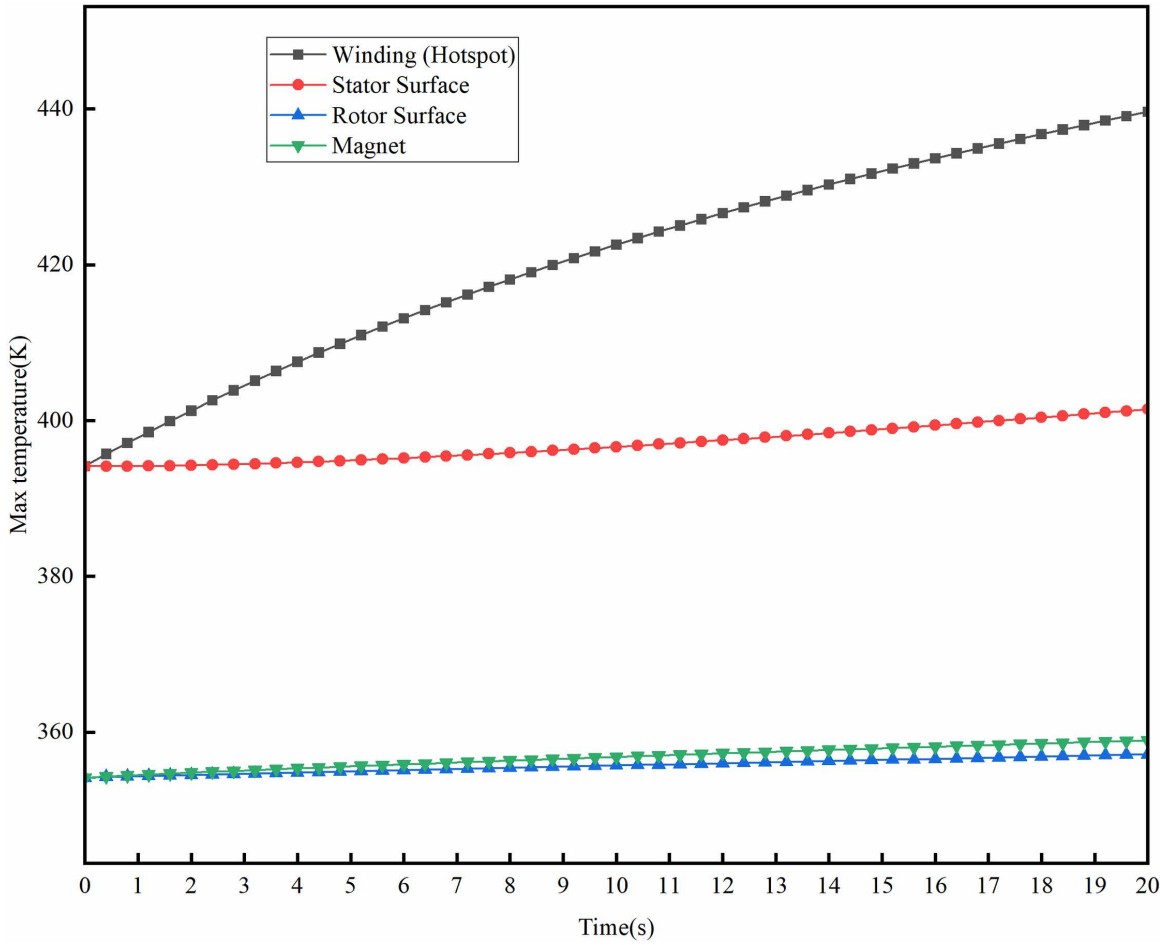

**Fig 17. Temperature distributions at maximum speed.**

To guarantee that the rotor's cooling structure exhibits substantial tensile strength under high-speed conditions, the stress distribution of the rotor and its corresponding cooling water circuit is examined at the motor's peak operational velocity. Fig 19 illustrates the stress distribution and deformation of the rotor. The maximum stress is 328MPa, with the chosen material B35AH300 demonstrating a tensile strength of 433MPa. The maximum deformation is 0.08 mm, which is relatively small and can be negligible.

## Conclusion

To address the cooling issue of the HPDPMSM, a frame-rotor dual-waterway cooling system has been devised. Parameters of waterway structures are established by thermodynamic calculations. Developed a simulation model to validate the cooling efficiency of the proposed cooling channel design. The influence of different canal configurations, coolants, water velocities, and operational conditions on the motor's heat dissipation capability is examined.

(1) An analysis of the cooling impact of the dual-waterway structure under rated working conditions is conducted. In comparison to the single-waterway structure, the rotor temperature rise is diminished by 47.3K, and the

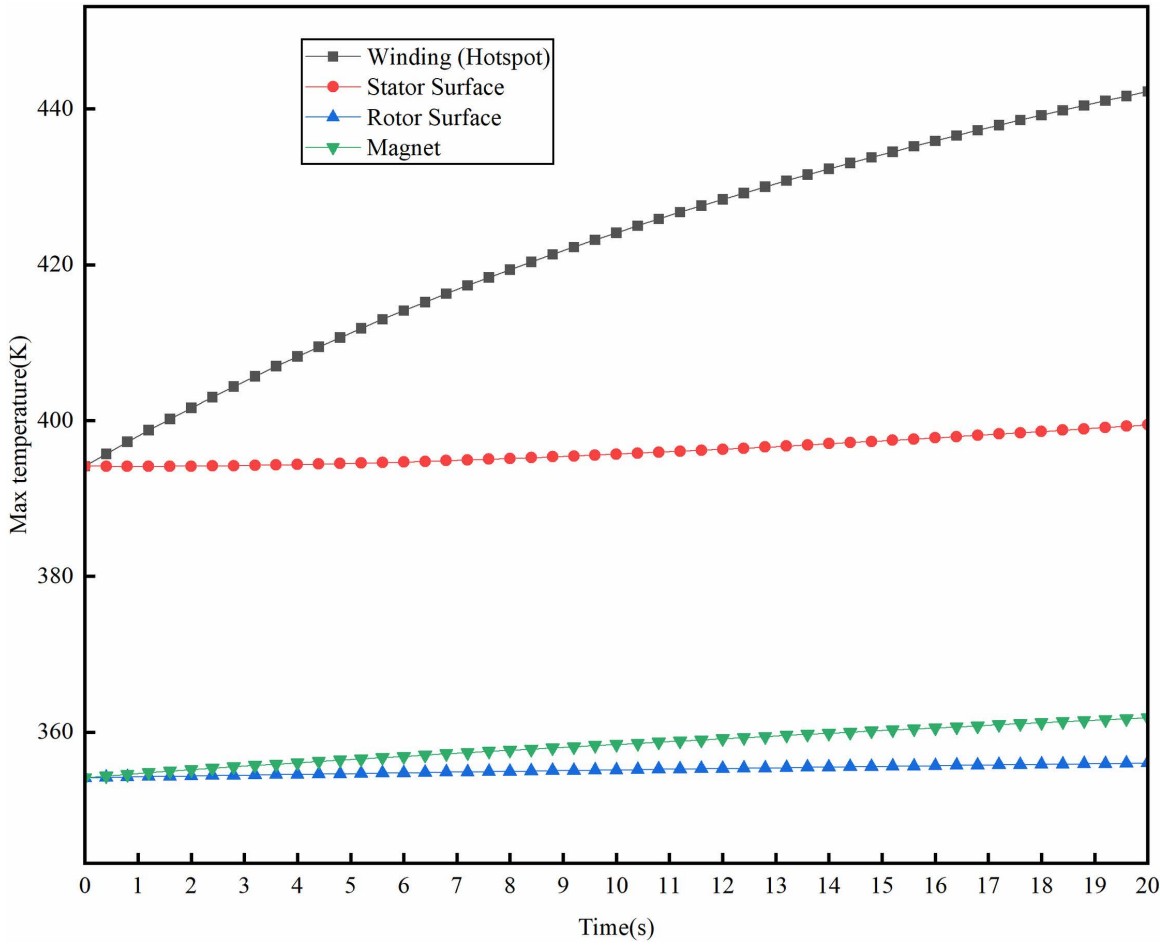

**Fig 18. Machine temperature distributions at peak power.**

winding temperature rise is decreased by 5.9K. The temperature field simulation of the motor operating at maximum speed and peak power meets the thermal rise requirements of H-class insulation and permanent magnet material.

(2) The impact of various coolants and flow rates on cooling efficiency is examined. Water offers the most efficient cooling. The impact of ethylene glycol solution varies slightly, as alterations in concentration diminish thermal conductivity and specific heat capacity. The cooling effect of oil is suboptimal. As velocity escalates, the motor's temperature elevation diminishes, however the water pressure grows linearly, with the best velocity determined to be 0.5m/s.

(3) The dual-waterway configuration can efficiently decrease rotor temperature and mitigate the risk of irreversible demagnetization of the permanent magnet at elevated temperatures. Simultaneously, the logic of the engineered waterway structure is examined and validated, yielding significant insights for the cooling system design of HPDPMSM powered by EVs.

A

Type: Equivalent (von-Mises) Stress
Unit: MPa
Time: 1

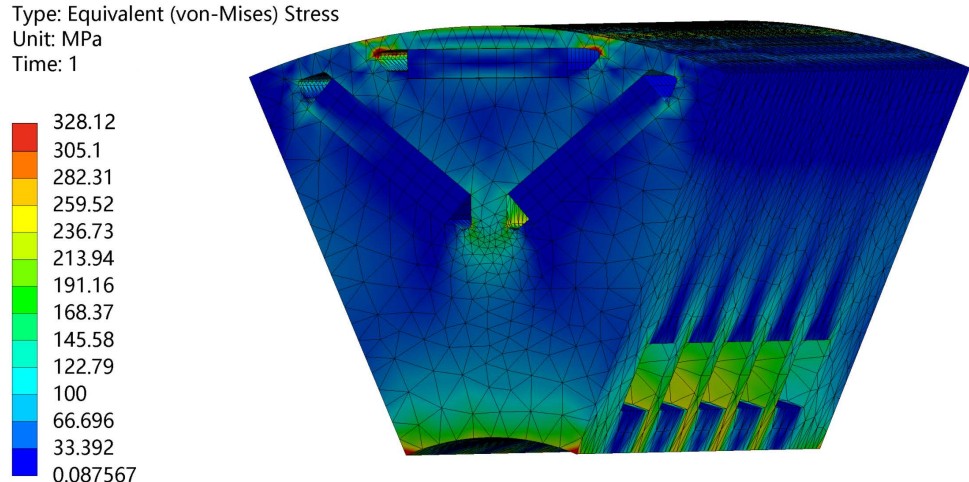
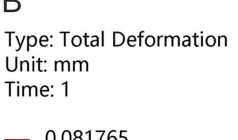

B

Type: Total Deformation
Unit: mm
Time: 1

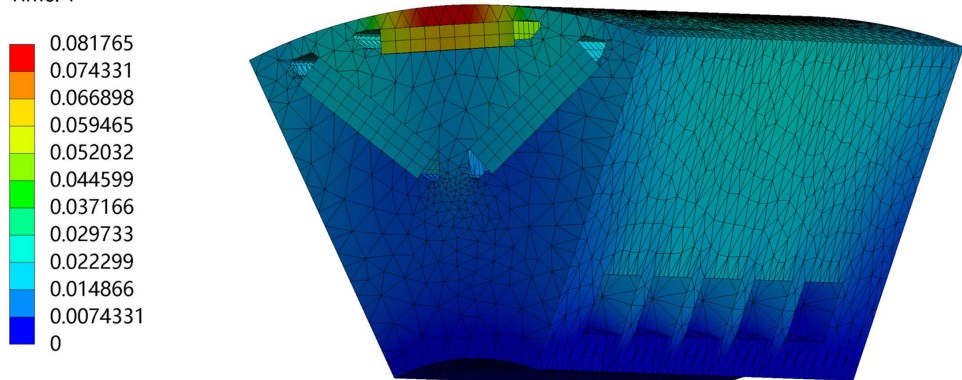

**Fig 19. Stress analysis of the rotor at the maximum speed. (A) Stress distribution, (B) Deformation.**

## Author contributions

**Conceptualization:** Yingying Xu, Mingxia XU.

**Funding acquisition:** Yingying Xu.

**Investigation:** Xiaoqian Duan, Aochen Han, Wenyu Yu.

**Methodology:** Xiaoqian Duan.

**Project administration:** Mingxia XU.

**Writing – original draft:** Xiaoqian Duan.

**Writing – review & editing:** Aochen Han.

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
