## [Decision Letter · Decision Letter 0]

9 Jul 2025

Dear Dr. XU,

Thank you for submitting your manuscript to PLOS ONE. After careful consideration, we feel that it has merit but does not fully meet PLOS ONE’s publication criteria as it currently stands. Therefore, we invite you to submit a revised version of the manuscript that addresses the points raised during the review process.

We look forward to receiving your revised manuscript.

Kind regards,

Xingwang Tang

Academic Editor

PLOS ONE

“The Fundamental Research Funds for the Provincial Universities of Liaoning, China, Grant number: LJ212410150005.”

“This research was funded by the Science and Technology Bureau of Dalian, Liaoning Province, China, Grant number: 2023RQ061.”

“The Fundamental Research Funds for the Provincial Universities of Liaoning, China, Grant number: LJ212410150005.”

6. We note that your Data Availability Statement is currently as follows: [All relevant data are within the manuscript and its Supporting Information files.]

7. PLOS requires an ORCID iD for the corresponding author in Editorial Manager on papers submitted after December 6th, 2016. Please ensure that you have an ORCID iD and that it is validated in Editorial Manager. To do this, go to ‘Update my Information’ (in the upper left-hand corner of the main menu), and click on the Fetch/Validate link next to the ORCID field. This will take you to the ORCID site and allow you to create a new iD or authenticate a pre-existing iD in Editorial Manager.

Reviewers' comments:

Reviewer's Responses to Questions

**Comments to the Author**

1. Is the manuscript technically sound, and do the data support the conclusions?

Reviewer #1: Yes

Reviewer #2: Partly

2. Has the statistical analysis been performed appropriately and rigorously?

Reviewer #1: Yes

Reviewer #2: I Don't Know

3. Have the authors made all data underlying the findings in their manuscript fully available?

Reviewer #1: Yes

Reviewer #2: No

4. Is the manuscript presented in an intelligible fashion and written in standard English?

Reviewer #1: Yes

Reviewer #2: Yes

Reviewer #1: Review Comments for the Paper on Dual-Channel Cooling System for High-Power-Density PMSM

The reviewer believes the paper can be accepted after minor revisions:

1. The authors need to clarify whether keywords should be included in the main text.

2. The introduction structure requires adjustment. The first and second paragraphs could be merged. Additionally, the authors should consider expanding references related to electric vehicles, such as fuel cell vehicles: https://www.sciencedirect.com/science/article/pii/S0306261921016718

https://www.sciencedirect.com/science/article/pii/S0019057825003040

3. The resolution of all figures should be improved, and vector graphics should be provided.

4. For Figure 13, is the "Maximum Temperature Rise" relative to ambient temperature or a specific baseline condition?

5. Key performance indicators for the "N48UH" permanent magnet material should be provided.

6. The English language requires further polishing.

Reviewer #2: This manuscript proposes an innovative frame-rotor dual water-circuit cooling system to solve the heat dissipation problem of high power density permanent magnet synchronous motors (HPDPMSM) in electric vehicles. The core is to form a circulation through the water channel outside the frame and the water channel inside the rotor support tube. The fluid-solid coupling simulation verifies that this structure can more effectively reduce the temperature rise of the motor compared to a single water channel, and analyzes the influence of different cooling media and flow rates. The proposed innovative architecture significantly improves the heat dissipation efficiency, solves the problem of rotor overheating in a targeted manner, and the structural design is innovative. However, the manuscript still has shortcomings. The research is only based on simulation and does not provide experimental verification, and does not deeply explore the potential engineering challenges of dual water channels in terms of sealing, complexity and cost of rotating parts.

The content of the manuscript is within the scope of the journal and can be of broad interest to readers. However, in terms of specific content, there is still room for improvement. Therefore, I decided to give the decision of major revision. It is recommended that the author properly absorb the reviewers' comments and make corresponding improvements and enhancements.

1. For the keywords, 'heat dissipation', 'thermodynamic calculation', 'temperature distribution', and 'motor structure' should be added to attract a broader readership.

2. The abstract and introduction do not clearly distinguish the innovation boundary between this study and previous works (for example, the dual-channel water cooling structure has been proposed in reference [14]), and it is necessary to quantify and compare the improvement ratio of the heat dissipation efficiency of the existing cooling system.

3. The fluid-solid coupled temperature field model has not been experimentally verified, and only single/double water channel structures are compared through simulation, which is not credible enough. It is recommended to increase test results, provide measured temperature data at key locations for comparison with simulation results, and verify the model error rate.

4. The quantitative effect of the physical parameters (viscosity, specific heat capacity) of different coolants (such as ethylene glycol aqueous solution, deionized water) on the heat dissipation performance is not explained, and only qualitative analysis is performed. It is recommended to clarify the mathematical basis for the selection of coolant.

5. Current research has not fully drawn on advanced thermal management theories in the field of energy and power, so the theoretical depth needs to be strengthened. It is recommended to refer to Song et al. (2025) for a review of the heat transfer mechanism of liquid cooling systems to clarify the principles of coordinated optimization of coolant physical parameters and flow channel design (10.1016/j.jpowsour.2025.237227). It is suggested to supplement the collaborative optimization equation of heat transfer coefficient and flow resistance.

6. The dynamic working condition verification method needs to be improved, especially the variable working condition thermal reliability analysis lacks a standardized test framework. The dynamic durability evaluation method of the vehicle power system proposed by Tang et al. (2024) can be transferred to this study, and it is recommended to supplement the high-speed transient temperature rise test (10.1109/TPEL.2024.3502499). It is necessary to quantify the thermal stability threshold of the cooling system during high-speed transient operation of the motor.

7. Dynamic operating condition analysis is incomplete. It only analyzes rated operating conditions and does not cover the robustness of the cooling system under high-speed or low-torque transient conditions. It is recommended to add temperature field simulation under variable speed and variable load conditions to verify the adaptability of the system under the full range of operating conditions.

8. Mechanical reliability analysis is missing. The centrifugal stress of the rotor spiral waterway under high-speed rotation and the durability of the sealing structure are not analyzed, which poses a safety hazard. It is recommended to add ANSYS structural mechanics simulation of the rotor waterway to calculate the strain at the maximum speed and the fatigue life of the seal.

9. Pure physical simulation is difficult to cover complex boundary conditions. As shown in [Energies 2024, 17(12), 3050], it is recommended to refer to the Transformer model and explore data-physics fusion simulation to improve the efficiency of temperature field prediction. Dynamic thermal management strategies can be trained using real-time monitoring data, reducing reliance on pure physical simulation.

**Do you want your identity to be public for this peer review?** For information about this choice, including consent withdrawal, please see our Privacy Policy

Reviewer #1: No

Reviewer #2: No

---

## [Author Response · Author response to Decision Letter 1]

14 Aug 2025

Dear Reviewers,

Hope you are doing well.

Thank you very much for your careful review and thoughtful evaluation of my paper. Your insightful comments and suggestions have been immensely valuable in guiding and enriching my research. Your contributions have played a crucial role in improving the quality of my work and advancing progress in the relevant field. We have made revisions to the article based on your feedback, and the relevant changes are highlighted.

The attachment is the author's revised draft and the revision notes. Please review it again.

---

## [Decision Letter · Decision Letter 1]

27 Aug 2025

Research on Dual-Waterway Cooling System of High-Power-Density Permanent Magnet Synchronous Machine

PONE-D-25-31613R1

Dear Dr. Xu,

We’re pleased to inform you that your manuscript has been judged scientifically suitable for publication and will be formally accepted for publication once it meets all outstanding technical requirements.

Kind regards,

Xingwang Tang

Academic Editor

PLOS ONE

Additional Editor Comments (optional):

Reviewers' comments:

Reviewer's Responses to Questions

**Comments to the Author**

Reviewer #1: All comments have been addressed

Reviewer #2: All comments have been addressed

2. Is the manuscript technically sound, and do the data support the conclusions?

Reviewer #1: Yes

Reviewer #2: Yes

3. Has the statistical analysis been performed appropriately and rigorously?

Reviewer #1: Yes

Reviewer #2: N/A

4. Have the authors made all data underlying the findings in their manuscript fully available?

Reviewer #1: Yes

Reviewer #2: Yes

5. Is the manuscript presented in an intelligible fashion and written in standard English?

Reviewer #1: Yes

Reviewer #2: Yes

Reviewer #1: The author has satisfactorily addressed all my concerns. The current manuscript is recommended for publication.

Reviewer #2: I have carefully reviewed and checked the updated version of the manuscript. I consider the issues proposed by the reviewers have been well addressed, and the current version of the manuscript can be accepted without further revisions.

**Do you want your identity to be public for this peer review?** For information about this choice, including consent withdrawal, please see our Privacy Policy

Reviewer #1: No

Reviewer #2: No

---

## [Editor Report · Acceptance letter]

PONE-D-25-31613R1

PLOS ONE

Dear Dr. XU,

I'm pleased to inform you that your manuscript has been deemed suitable for publication in PLOS ONE. Congratulations! Your manuscript is now being handed over to our production team.

Kind regards,

on behalf of

Dr. Xingwang Tang

Academic Editor

PLOS ONE